# Detecting Textual Adversarial Examples through Randomized Substitution and Vote

**Xiaosen Wang**[*]                **Yifeng Xiong**[*]                **Kun He**[†]

School of Computer Science and Technology, Huazhong University of Science and Technology, Wuhan, China
{xiaosen,xiongyf,brooklet60}@hust.edu.cn

## Abstract

A line of work has shown that natural text processing models are vulnerable to adversarial examples. Correspondingly, various defense methods are proposed to mitigate the threat of textual adversarial examples, *e.g*., adversarial training, input transformations, detection, *etc*. In this work, we treat the optimization process for synonym substitution based textual adversarial attacks as a specific sequence of word replacement, in which each word mutually influences other words. We identify that we could destroy such mutual interaction and eliminate the adversarial perturbation by randomly substituting a word with its synonyms. Based on this observation, we propose a novel textual adversarial example detection method, termed *Randomized Substitution and Vote* (RS&V), which votes the prediction label by accumulating the logits of $k$ samples generated by randomly substituting the words in the input text with synonyms. The proposed RS&V is generally applicable to any existing neural networks without modification on the architecture or extra training, and it is orthogonal to prior work on making the classification network itself more robust. Empirical evaluations on three benchmark datasets demonstrate that our RS&V could detect the textual adversarial examples more successfully than the existing detection methods while maintaining the high classification accuracy on benign samples.

## 1 INTRODUCTION

Deep Neural Networks (DNNs) are known to be vulnerable to adversarial examples [Szegedy et al., 2014, Goodfellow et al., 2015, Alzantot et al., 2018, Wang and He, 2021], in which human-imperceptible modifications on the benign samples could mislead the model prediction. More seriously, adversarial examples have been found in a variety of deep learning tasks, including Computer Vision (CV) [Goodfellow et al., 2015, Madry et al., 2018] and Natural Language Processing (NLP) [Papernot et al., 2016, Alzantot et al., 2018], leading to great threats to the security of numerous real-world applications, *e.g*., spam filtering, malware detection, *etc*. Consequently, it has gained broad attention on generating or defending adversarial examples for various deep learning tasks.

In the field of NLP, numerous adversarial attack methods have been proposed recently, which could be roughly split into three categories, namely character-level attacks [Gao et al., 2018, Ebrahimi et al., 2018, Li et al., 2019], word-level attacks [Papernot et al., 2016, Alzantot et al., 2018, Ren et al., 2019, Garg and Ramakrishnan, 2020, Zang et al., 2020, Wang et al., 2021b, Maheshwary et al., 2021], and sentence-level attacks [Liang et al., 2018, Ribeiro et al., 2018, Zhang et al., 2019, Wang et al., 2020]. Among these methods, synonym substitution based attacks that belong to word-level attacks, are widely investigated because they naturally satisfy the lexical, grammatical and semantic constraints of natural texts. In this work, we focus on effectively detecting the textual adversarial examples generated by synonym substitution based attacks.

To mitigate the threat of textual adversarial examples, some researchers propose to enhance the model to resist adversarial examples, such as input pre-processing [Wang et al., 2021a, Zhou et al., 2021], adversarial training [Wang et al., 2021b, Dong et al., 2021], certified defense [Jia et al., 2019, Huang et al., 2019, Shi et al., 2020], *etc*. Meanwhile, another line of work [Zhou et al., 2019, Mozes et al., 2021] focuses on detecting the adversarial examples so that we could further reject the detected adversarial examples or recover the benign samples before feeding them into the models. Detection methods usually utilize an extra detection module that is generally applicable to any target model and is very

---

[*]Equal Contribution
[†]Corresponding author.

*Accepted for the 38*[th] *Conference on Uncertainty in Artificial Intelligence*  (UAI 2022).

popular in industrial applications [Lu et al., 2017] due to the minor decay on clean accuracy. Currently, however, there exists little attention on textual adversarial example detection. Moreover, the two lines of defense are complementary to each other and could be integrated together to further enhance the model robustness.

In this work, we aim to improve the adversarial example detection accuracy for text classification tasks against synonym substitution based attacks. We treat the optimization process for synonym substitution based attacks as a specific sequence for word replacement, denoted as replacement sequence, in which each word mutually influences other words. Inspired by the existing input pre-processing based defenses [Wang et al., 2021a, Zhou et al., 2021], we observe that randomized synonym substitution could destroy such mutual interaction and eliminate the adversarial perturbation with high probability. To this end, we propose a novel textual adversarial example detection method, termed *Randomized Substitution and Vote* (RS&V). Specifically, given an input text $x$, we first generate a set of samples $\{x_1, x_2, \cdots, x_k\}$ by randomly substituting some words $w_i \in x$ with their arbitrary synonyms $\hat{w}_i$. Then we accumulate the logits output of the $k$ processed samples and treat $x$ as an adversarial example if the prediction for $x$ is inconsistent with the prediction voted by the $k$ samples.

Our main contributions are as follows:

- We identify that randomized synonym substitution could destroy the mutual interaction among the words in the replacement sequence.
- We propose a simple yet effective adversarial example detection method RS&V against synonym substitution based attack, which is rarely investigated but very significant in real-world applications.
- Empirical evaluations demonstrate that RS&V outperforms the state-of-the-art baselines no matter whether the attackers could access the detection module and maintain the high accuracy on benign samples.
- RS&V utilizes the model generalization and robustness to randomized synonym substitution to detect the adversarial examples without any additional training or modification on the architecture and is generally applicable to any existing neural networks.

## 2 RELATED WORK

Recently, adversarial examples for text classification tasks, *i.e.* textual adversarial examples, have attracted great interest, especially on textual adversarial attacks [Papernot et al., 2016, Ebrahimi et al., 2018, Ribeiro et al., 2018, Alzantot et al., 2018, Ren et al., 2019, Zang et al., 2020], followed by an increasing attention on textual adversarial defenses [Jia et al., 2019, Liu et al., 2020, Jones et al., 2020, Wang et al., 2021b, Mozes et al., 2021, Yang et al., 2022].

**Textual Adversarial Attacks.** According to the different strategies to generate adversarial examples, we could roughly divide the existing textual adversarial attacks into three categories: 1) Character-level attacks usually swap, flip, remove or insert characters in the input text to generate the adversary, *e.g.*, DeepWordBug [Gao et al., 2018], HotFlip [Ebrahimi et al., 2018], TextBugger [Li et al., 2019]. 2) Word-level attacks replace the words with other unrelated words in the dictionary [Papernot et al., 2016] or only substitute the words with its synonyms [Alzantot et al., 2018]. The latter kind of word-level attacks could preserve the semantic consistency and have been widely accepted [Alzantot et al., 2018, Ren et al., 2019, Li et al., 2020, Zang et al., 2020, Jin et al., 2020, Wang et al., 2021a, Maheshwary et al., 2021, Guo et al., 2021, Yuan et al., 2021]. 3) Sentence-level attacks insert a parenthesis [Liang et al., 2018, Wang et al., 2020] or rephrase the original sentence without changing the original meaning [Iyyer et al., 2018, Zhang et al., 2019]. Some works [Liang et al., 2018, Li et al., 2019] also utilize both character-level and word-level attacks to achieve better attack performance.

**Textual Adversarial Defenses.** To mitigate the threat of textual adversarial examples, various defense methods have also been proposed, such as spell checker [Pruthi et al., 2019], input pre-processing [Wang et al., 2021a, Zhou et al., 2021], adversarial training [Wang et al., 2021b, Dong et al., 2021, Si et al., 2021], certified defense [Jia et al., 2019, Huang et al., 2019, Shi et al., 2020] and adversarial example detection [Zhou et al., 2019, Mozes et al., 2021]. For the detection methods, Zhou et al. [2019] propose *Learning to Discriminate Perturbation* (DISP), which trains a perturbation discriminator and embedding estimator to detect the adversarial examples and recover the benign samples. Mozes et al. [2021] propose *Frequency-Guided Word Substitutions* (FGWS) that substitutes all the low frequency words in the input text with the most frequent synonyms to eliminate the adversarial perturbation. Compared with the defense methods that enhance the model robustness, detection methods usually adopt an extra detection module which leads to little decay on clean accuracy, making it popular in real-world applications.

In this work, we propose a simple yet effective method, called Randomized Substitution and Vote (RS&V), to detect the synonym substitution based textual adversarial examples without modifying the architecture or training process, which is generally applicable to any model.

## 3 METHODOLOGY

In this section, we first introduce the preliminary of textual adversarial attacks and our motivation. Then we give a detailed description on the proposed Randomized Substitution and Vote (RS&V) method. Code is available at `https://github.com/JHL-HUST/RSV`.

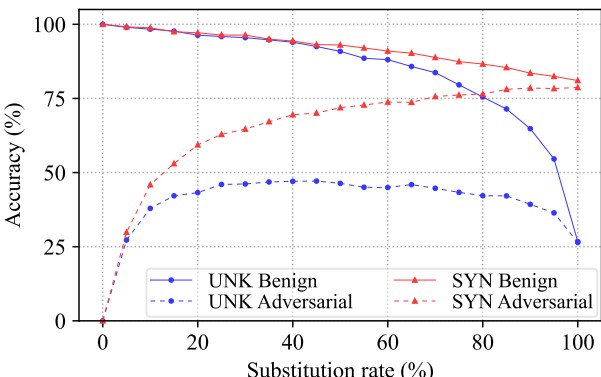

Figure 1: The classification accuracy (%) on benign samples and the corresponding adversarial examples generated by Textfooler by randomly substituting the words with "UNK" token (blue line) or synonyms (red line) for different rates. The results are evaluated on 1,000 correctly classified samples from AG's News test set for Word-CNN.

## 3.1 PRELIMINARY

Let $\mathcal{X}$ be the set of input texts and $\mathcal{Y} = \{y_1, y_2, \cdots, y_N\}$ be the set of corresponding labels. Given an input text $x = \langle w_1, w_2, \cdots, w_n \rangle \in \mathcal{X}$ with the ground-truth label $y_{true} \in \mathcal{Y}$, a well-trained natural language model $f : \mathcal{X} \to \mathcal{Y}$ predicts its label $\bar{y} = y_{true}$ with high probability, using the maximum posterior probability:

$$\bar{y} = \arg\max_{y_i \in \mathcal{Y}} f(y_i | x). \tag{1}$$

The attacker perturbs the natural text $x$ slightly to generate a textual adversarial example $x^{adv}$ which misleads the model prediction:

$$\arg\max_{y_i \in \mathcal{Y}} f(y_i | x^{adv}) \neq y_{true}. \tag{2}$$

In this work, we focus on detecting the synonym substitution based adversarial examples generated by substituting a few words, in which each word $w_i \in x$ is substituted with one of its synonym $\hat{w}_i$, so as to maintain the semantic consistency while attacking.

For the task of adversarial example detection, given an input text $x$, the detector $D$ should determine whether $x$ is benign sample or adversarial example. A stronger detector would also contain a corrector $E$, which could restore the correct label for the input adversarial example:

$$\bar{y}_{restore} = \begin{cases} \bar{y} & \text{if } D(x) = \text{False}; \\ E(x) & \text{if } D(x) = \text{True}. \end{cases} \tag{3}$$

## 3.2 MOTIVATION

The optimization process of synonym substitution based textual adversarial attacks could be regarded as searching

a specific sequence for word replacement, termed replacement sequence, in which the words mutually influence each other and contribute together to mislead the target classifier. Intuitively, we postulate that we could eliminate the perturbation if we could successfully break the mutual interaction of the words in the replacement sequence.

To validate our hypothesis, we first generate adversarial examples using the Textfooler attack [Jin et al., 2020] on 1,000 randomly sampled texts from the AG's News test set [Zhang et al., 2015] that are correctly classified by Word-CNN [Kim, 2014]. To break the interaction among words in the replacement sequence, we first randomly mask words in the benign samples or adversarial examples as unknown ("UNK") token, denoted as *UNK Benign* or *UNK Adversarial*, with different rates and feed these processed texts to the model. As shown in Figure 1, the classification accuracy is still over 90% when we mask 40% words in the benign text, showing the stability and robustness of model to such randomized mask. In contrast, the classification accuracy on adversarial examples increases remarkably when we mask more words and we could recover 50% adversarial examples when masking 40% words. This validates our hypothesis that we could eliminate the adversarial perturbation if we successfully break the mutual interaction of words in the replacement sequence.

However, since we randomly substitute the meaningful words with meaningless "UNK" token, the semantic meaning of the text would be destroyed significantly when masking a certain number of words, leading to poor performance on benign samples and limited improvement on adversarial examples. Inspired by the existing synonym substitution based attacks, we attempt to randomly substitute the words with its synonyms to break the interaction of words without destroying the original semantic meaning. As shown in Figure 1, we find that randomly substituting words with its synonyms in the benign samples or adversarial examples, denoted as *SYN Benign* or *SYN Adversarial*, could consistently and significantly improve the robust accuracy against adversarial examples at the same time maintaining the high accuracy on benign samples under various substitution rates, which further validates our hypothesis.

Based on the above observation, we propose a novel textual adversarial example detection method called RS&V, which randomly substitutes the words in the input text with their synonyms to effectively detect the adversarial examples and restore the correct label for the input adversarial example.

## 3.3 RANDOMIZED SUBSTITUTION AND VOTE

Motivated by the observation that randomly substituting the words with their synonyms could eliminate the adversarial perturbation remarkably while maintaining the high clean accuracy, we propose a novel adversarial example detection

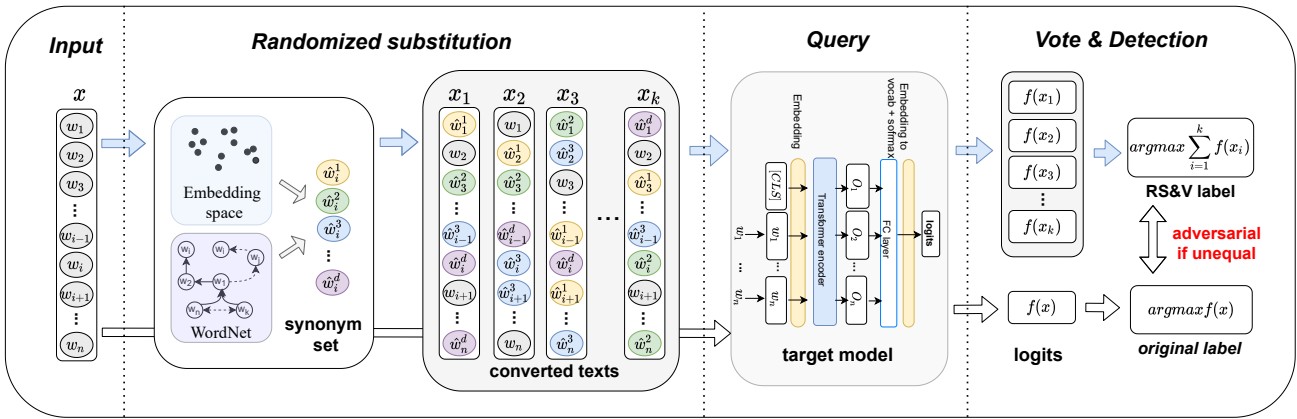

Figure 2: The overall framework of the proposed RS&V method. For an input $x$, we first generate the synonym set $\mathcal{S}(w_i)$ for each word $w_i \in x$ using the GloVe vector as well as WordNet, and randomly substitute a word $w_i \in x$ with its arbitrary synonym $\hat{w}_i^j \in \mathcal{S}(w_i)$ to generate multiple texts. Then we feed these generated texts and accumulate the logits to get the voted label. Finally, we regard $x$ as an adversarial example if the RS&V label is not consistent with the prediction label for $x$.

---

**Algorithm 1** The RS&V Algorithm

---

**Input:** Input text $x = \{w_1, w_2, ..., w_n\}$, target classifier $f$, substitution rate $p$, number of votes $k$, stopword selection portion $s$.

**Output:** Detection result and restored label

1: Calculate the stopword set $\mathcal{W}$ containing the top $s$ high frequency words in the training set
2: Initialize converted text set $\mathcal{X} = \emptyset$
3: **for** $i = 1 \to k$ **do**   ▷ Randomized Substitution
4:  Initialize a new text $x_i = x$
5:  Randomly sample $n \cdot p$ words for $\mathcal{P}$ from $x_i / \mathcal{W}$
6:  **for** each word $w_t \in \mathcal{P}$ **do**
7:    Randomly select a synonym $\hat{w}_t^j \in \mathcal{S}(w_t)$
8:    Substitute $w_t \in x_i$ with $\hat{w}_t^j$
9:  **end for**
10:  $\mathcal{X} = \mathcal{X} \cup x_i$
11: **end for**
12: Calculate the prediction label for input text $x$: $\bar{y} = \operatorname{argmax} f(x)$   ▷ Vote & Detection
13: Calculate the voted label: $\bar{y}_v = \operatorname{argmax} \sum_{i=1}^{k} f(x_i)$
14: **if** $\bar{y} = \bar{y}_v$ **then**
15:  **return** False, $\bar{y}$      ▷ Benign sample
16: **end if**
17: **return** True, $\bar{y}_v$     ▷ Adversarial example

---

method called Randomized Substitution and Vote (RS&V), to detect textual adversarial examples. As shown in Figure 2, given an input text $x$, RS&V contains two main stages, *i.e.* randomized substitution and vote & detection, for detecting whether $x$ is an adversarial example and restoring its correct label if it is adversarial.

**Randomized Substitution.** Given an input text $x = \langle w_1, w_2, \cdots w_n \rangle$, we first construct the synonym set $\mathcal{S}(w_i) = \{\hat{w}_i^1, \hat{w}_i^2, \cdots, \hat{w}_i^k\}$ for each word $w_i \in x$ by

combining the synonyms from WordNet [Miller, 1992] and neighbor words within a given Euclidean distance in the GloVe embedding space post-processed by counter-fitting [Mrkšić et al., 2016]. Then we randomly sample $n \cdot p$ words $\{w_i \in x\}$ and substitute each selected word with its arbitrary synonym $\hat{w}_i^j \in \mathcal{S}(w_i)$ to break the interaction among words in the replacement sequence for adversarial examples. Moreover, we find that substituting some high-frequency words (*e.g.*, "a", "the", "hello", *etc.*) is not only useless to mitigate the adversarial effect, but also has negative impact on classifying benign samples. Inspired by the Term Frequency–Inverse Document Frequency (TF-IDF) technique, we set the words with the top $s$ frequency among the training set as stopwords, where $s$ is a hyper-parameter, and ignore these words for randomized sampling. Such randomized substitution would repeat for $k$ times to generate $k$ different converted texts $\{x_1, x_2, \cdots, x_k\}$.

**Vote & Detection.** Although we have shown that randomized substitution could remarkably mitigate the adversarial effect with an acceptable decay on the clean accuracy, we seek to further improve the recovery ability of randomized substitution and decrease the negative impact on benign samples. To eliminate the variance introduced by randomized substitution and stabilize the detection, we feed $k$ various converted texts $\{x_1, x_2, \cdots, x_k\}$ and accumulate the logits output on these texts to vote for the final label $\operatorname{argmax} \sum_{i=1}^{k} f(x_i)$. Note that the method is different from the naive approach depicted in Section 3.2. Our RS&V would detect the input text $x$ as an adversarial example if the voted RS&V label is inconsistent with the prediction label $\operatorname{argmax} f(x)$ and output the RS&V label as the prediction result for $x$.

In summary, as depicted in Figure 2, the proposed RS&V first generates multiple samples by randomly substituting words in text with synonyms and accumulates the logits of

| Dataset | Task | # Classes | # Average words | # Training samples | # Testing samples |
|---|---|---|---|---|---|
| AG's News | News categorization | 4 | 38 | 120,000 | 7,600 |
| IMDB | Sentiment analysis | 2 | 227 | 25,000 | 25,000 |
| Yahoo! Answers | Topic classification | 10 | 32 | 1,400,000 | 60,000 |

Table 1: Statistics on the datasets. # indicates the number and # Average words indicates the average number of words per sample text in the dataset.

these samples to vote the final label. If the voted label is not consistent with the prediction label for the input text, RS&V would treat the input text as adversarial example and output the voted label. The overall algorithm of RS&V is summarized in Algorithm 1.

# 4 EXPERIMENTS

To validate the effectiveness of the proposed RS&V, we conduct extensive evaluations on three benchmark datasets for three models against five popular adversarial attacks. In this section, we first specify the experimental setup, then we compare the detection performance of RS&V with two detection baselines, and demonstrate that RS&V could achieve better detection performance in two settings and maintain the high accuracy on benign samples. Finally, we investigate the impact of three hyper-parameters, *i.e.* substitution rate $p$, number of votes $k$, and stopword selection portion $s$.

## 4.1 EXPERIMENTAL SETUP

**Datasets and Models.** We adopt three widely investigated benchmark datasets, *i.e.* IMDB [Maas et al., 2011], AG's News, Yahoo! Answers [Zhang et al., 2015], including sentiment analysis and news or topic classification. Details about the datasets are summarized in Table 1. To validate the generalization to model architectures of our RS&V, we adopt several popular deep learning models that exhibit state-of-the-art performance on text classification tasks, including Word-CNN [Kim, 2014] and two pre-trained language models, BERT [Devlin et al., 2019] and RoBERTa [Liu et al., 2019].

**Evaluation Setup.** We consider static attack evaluation (SAE) and targeted attack evaluation (TAE) [Si et al., 2021]. In SAE setting, the attacker does not know the detection module and generates adversarial examples on the original model. In TAE setting, the attacker could access and attack the detection module simultaneously when attacking target model. We evaluate the detection performance against various attacks, *i.e.* GA [Alzantot et al., 2018], PWWS [Ren et al., 2019], PSO [Zang et al., 2020], Textfooler [Jin et al., 2020] and HLA [Maheshwary et al., 2021], using F1 score and detection accuracy, *i.e.* classification accuracy on samples recovered by detection modules.

**Baselines.** We compare our method with normally trained models and two recently proposed detection methods designed for synonym substitution based attacks:

- DISP [Zhou et al., 2019] trains a perturbation discriminator to identify the perturbed tokens and an embedding estimator to reconstruct the original text.
- FGWS [Mozes et al., 2021] replaces the low-frequency words with their most frequent synonyms in the dictionary to eliminate the adversarial perturbation.

**Hyperparameters.** For our RS&V, the number of votes is set to 25, the substitution rate $p$ is set to $80\%$ on IMDB and $60\%$ on AG's News as well as Yahoo! Answers based on the text length, and the stopword selection portion $s$ is set to $2\%$. We mix up the synonyms from WordNet [Miller, 1992] and embedding space as synonyms set, where the embedding space synonyms were calculated by ranking the Euclidean distance between words based on GloVe vectors processed by counter-fitting [Mrkšić et al., 2016]. We select at most 6 words on embedding space to keep the diversity of the synonyms set, and the portion $s$ for stopwords is set to 5 for IMDB and 2 for AG's News and Yahoo base on the frequency drops, For the baselines, we use the same way in the original paper to tune hyper-parameters on the validation set.

## 4.2 EVALUATION IN THE SAE SETTING

To validate the effectiveness of RS&V, we evaluate the detection accuracy, the classification accuracy on the recovered samples by the detection methods and F1 score against adversarial attacks on three datasets, and do comparison with DISP and FGWS. Due to the high computational cost of generating adversarial examples for the attack methods, we randomly select 1,000 samples on each dataset and adopt the attacks to generate adversarial examples for each model. The results are summarized in Table 2.

We could observe that RS&V generally exhibits superior detection performance against various adversarial attacks with high classification accuracy on benign samples. Specifically, all the normally trained models achieve the lowest classification accuracy on three datasets against five attacks. Compared with two detection baselines, RS&V consistently achieve much better detection accuracy and F1 score on three datasets for various models against five adversarial

| Dataset | Model | Method | Clean | GA | | PWWS | | PSO | | Textfooler | | HLA | | Average | |
|---|---|---|---|---|---|---|---|---|---|---|---|---|---|---|---|
| | | | | Acc. | F1 | Acc. | F1 | Acc. | F1 | Acc. | F1 | Acc. | F1 | Acc. | F1 |
| AG's News | CNN | N/A | 92.1 | 43.6 | – | 37.1 | – | 36.4 | – | 24.8 | – | 41.9 | – | 36.8 | – |
| | | DISP | 91.6 | 77.3 | 83.7 | 76.5 | 85.0 | 78.0 | 85.8 | 69.0 | 80.8 | 79.2 | 85.9 | 76.0 | 84.2 |
| | | FGWS | 91.3 | 76.2 | 80.3 | 75.8 | 83.4 | 76.2 | 84.0 | 77.5 | 88.3 | 80.0 | 85.8 | 77.1 | 84.4 |
| | | RS&V | 91.3 | **84.1** | **90.2** | **85.1** | **92.2** | **86.8** | **94.0** | **86.0** | **94.8** | **88.1** | **94.8** | **86.0** | **93.2** |
| | BERT | N/A | 94.9 | 68.5 | – | 74.9 | – | 59.2 | – | 61.3 | – | 62.0 | – | 65.2 | – |
| | | DISP | 94.5 | 85.3 | 77.8 | 85.2 | 70.3 | 83.8 | 80.9 | 83.0 | 81.1 | 85.6 | 84.2 | 84.6 | 78.9 |
| | | FGWS | 94.6 | 87.2 | 82.9 | 88.8 | 82.5 | 87.7 | 88.0 | 89.2 | 91.5 | 89.1 | 90.5 | 88.4 | 87.1 |
| | | RS&V | 94.6 | **90.5** | **90.4** | **91.3** | **88.8** | **91.7** | **94.2** | **92.5** | **96.2** | **92.3** | **90.8** | **92.8** | **92.1** |
| | RoBERTa | N/A | 93.5 | 71.9 | – | 78.7 | – | 66.8 | – | 74.6 | – | 68.8 | – | 72.2 | – |
| | | DISP | 93.4 | 84.8 | 75.8 | 84.8 | 65.3 | 85.9 | 84.9 | 82.8 | 67.3 | 86.8 | 85.3 | 85.0 | 75.7 |
| | | FGWS | 93.1 | 86.2 | 80.0 | 87.2 | 74.6 | 87.0 | 86.6 | 88.5 | 85.2 | 88.3 | 88.9 | 87.4 | 83.1 |
| | | RS&V | 93.4 | **89.6** | **88.9** | **90.3** | **87.0** | **91.5** | **94.7** | **91.2** | **92.6** | **91.5** | **94.7** | **90.8** | **91.6** |
| IMDB | CNN | N/A | 87.2 | 6.2 | – | 1.5 | – | 2.7 | – | 0.6 | – | 17.4 | – | 5.7 | – |
| | | DISP | 87.2 | 48.8 | 68.3 | 43.1 | 64.8 | 53.3 | 74.4 | 39.3 | 61.2 | 62.0 | 77.4 | 49.3 | 69.2 |
| | | FGWS | 86.5 | 64.7 | 82.8 | 64.7 | 84.0 | 69.7 | 87.5 | 72.6 | 90.0 | 72.8 | 87.4 | 68.9 | 86.3 |
| | | RS&V | 86.3 | **79.6** | **94.0** | **80.2** | **94.8** | **80.9** | **95.0** | **79.2** | **94.1** | **81.7** | **94.5** | **80.3** | **94.5** |
| | BERT | N/A | 91.9 | 15.4 | – | 26.7 | – | 5.6 | – | 9.5 | – | 15.7 | – | 14.6 | – |
| | | DISP | 91.8 | 64.3 | 77.5 | 63.7 | 72.2 | 68.7 | 84.1 | 62.0 | 77.6 | 74.5 | 86.7 | 66.6 | 79.6 |
| | | FGWS | 92.5 | 80.6 | 90.9 | 79.5 | 88.2 | 82.0 | 92.9 | 83.0 | 93.2 | 84.8 | 94.0 | 82.0 | 91.8 |
| | | RS&V | 92.1 | **87.8** | **96.0** | **88.2** | **95.4** | **88.5** | **96.9** | **89.1** | **97.2** | **89.9** | **97.2** | **88.7** | **96.5** |
| | RoBERTa | N/A | 94.2 | 18.3 | – | 29.9 | – | 7.0 | – | 34.3 | – | 21.8 | – | 22.3 | – |
| | | DISP | 93.9 | 66.3 | 77.4 | 64.1 | 70.0 | 67.4 | 81.7 | 68.7 | 73.3 | 76.7 | 86.1 | 68.6 | 77.7 |
| | | FGWS | 94.4 | 81.0 | 90.1 | 82.0 | 89.1 | 83.2 | 92.9 | 86.6 | 92.7 | 85.7 | 93.4 | 83.7 | 91.6 |
| | | RS&V | 94.6 | **89.4** | **95.9** | **88.8** | **94.7** | **91.0** | **97.4** | **91.4** | **96.5** | **90.8** | **96.9** | **90.3** | **96.3** |
| Yahoo! Answers | CNN | N/A | 69.5 | 4.7 | – | 5.6 | – | 2.6 | – | 3.9 | – | 4.3 | – | 4.2 | – |
| | | DISP | 69.8 | 37.4 | 67.1 | 35.6 | 63.8 | 39.3 | 70.6 | 35.9 | 66.5 | 45.0 | 76.3 | 38.6 | 68.9 |
| | | FGWS | 68.0 | 49.7 | 82.6 | 48.2 | 80.9 | 49.4 | 82.2 | 40.6 | 72.1 | 39.9 | 75.0 | 45.6 | 78.6 |
| | | RS&V | 69.3 | **63.0** | **92.6** | **62.1** | **92.8** | **63.2** | **93.3** | **61.6** | **93.2** | **62.6** | **91.9** | **62.5** | **92.8** |
| | BERT | N/A | 76.7 | 13.8 | – | 25.6 | – | 8.9 | – | 17.9 | – | 11.5 | – | 15.5 | – |
| | | DISP | 76.7 | 50.0 | 74.5 | 50.5 | 68.6 | 53.8 | 80.4 | 51.7 | 74.8 | 56.0 | 81.9 | 52.4 | 76 |
| | | FGWS | 75.7 | 62.2 | 88.0 | 62.7 | 85.9 | 62.4 | 88.7 | 66.0 | 90.4 | 65.5 | 91.1 | 63.8 | 88.8 |
| | | RS&V | 75.8 | **67.4** | **92.3** | **68.7** | **91.0** | **69.7** | **93.7** | **71.6** | **94.1** | **70.0** | **93.9** | **69.5** | **93.0** |
| | RoBERTa | N/A | 74.7 | 19.8 | – | 33.7 | – | 15.2 | – | 41.7 | – | 19.6 | – | 26.0 | – |
| | | DISP | 74.7 | 48.0 | 68.7 | 50.4 | 61.3 | 50.9 | 75.1 | 53.7 | 57.6 | 55.2 | 78.6 | 51.7 | 68.3 |
| | | FGWS | 74.8 | 62.3 | 87.5 | 64.9 | 85.9 | 63.3 | 88.5 | 67.2 | 86.3 | 65.7 | 90.1 | 64.7 | 87.7 |
| | | RS&V | 76.0 | **66.4** | **90.3** | **66.8** | **86.7** | **68.1** | **92.2** | **68.3** | **86.8** | **68.7** | **92.8** | **67.7** | **89.8** |

Table 2: The classification accuracy (%) and F1 score (%) of various detection methods for Word-CNN and BERT on AG's News, IMDB and Yahoo! Answers. N/A denotes the normally trained model without the detection module.

attacks, while maintaining the high classification accuracy on benign samples. For instance, on Yahoo! Answers dataset for RoBERTa model, RS&V exhibits even better (1.3% higher) classification accuracy on benign samples than normally trained model and averagely outperforms DISP and FGWS with a clear margin of 16.1% and 3.0%, respectively. This convincingly validates the high effectiveness of RS&V.

To further gain insight on the performance improvement of RS&V, we present a randomly sampled adversarial example for Word-CNN from AG's News generated by Textfooler and the recovered examples for three detection methods in Table 3. As we can see, none of DISP, FGWS, and RS&V could really recover the benign sample as we expect, but all of them could lead to correct classification result. The results also show the fragility of textual adversarial example and might inspire more powerful detection and defense methods by pre-processing the input text without extra training or modification on the architecture in the future.

## 4.3 EVALUATION IN TAE SETTING

The SAE setting, where the attacker does not know there is a detection module, is the only case considered by the existing textual adversarial example detection methods. To

| Method | Text | Prediction / Confidence |
|---|---|---|
| Adv. | Tabbed **searches mistakes** (Browsing Flaws) Detected. Tabbed **searches** (Browsing), one of the more popular **hallmarks** (features) built into **other network** (alternative Web) browsers, contains a security **weakness** (flaw) that puts users at risk of spoofing attacks, **investigation** (research) firm Secunia **warn** (warned) on Wednesday. | World / 54.2% (Sci/Tec / 99.0%) |
| DISP | Tabbed searches mistakes Detected. Tabbed searches, one of the more popular hallmarks built into **the** network browsers, contains a security weakness that puts users at risk of spoofing attacks, investigation firm Secunia **said** on Wednesday. | Sci/Tec / 56.6% |
| FGWS | Tabbed searches mistakes Detected. Tabbed searches, one of the more popular **features** built into other network browsers, contains a security weakness that puts users at risk of **spoof** attacks, investigation firm Secunia warn on Wednesday. | Sci/Tec / 61.5% |
| RS&V | Tabbed **seek** mistakes Detected. Tabbed searches, one of the more popular **features** built into other network browsers, **involves** a security **flaw** that **blue** users at risk of spoofing attacks, investigation firm Secunia warn on Wednesday. | Sci/Tec / 93.6% |
| | Tabbed searches **errors** Detected. Tabbed **look**, one of the more popular hallmarks **establish** into other network **browser**, contains a security weakness that **poses** users at risk of spoofing attacks, investigation firm Secunia warn on Wednesday. | |
| | Tabbed **explore** mistakes **detecting**. Tabbed **searching**, one of the more popular **authentication** built into other network browsers, **involves** a security **flaw** that puts users at risk of spoofing attacks, investigation firm Secunia warn on Wednesday. | |

Table 3: The adversarial example generated by Textfooler and its corresponding samples used in RS&V and two detection baselines for Word-CNN on AG's News. We highlight the words replaced by Textfooler in **Red** and the words replaced by the detection methods in **Blue**. The words in the parentheses are the words in the original input text, which could be correctly classified. We also mark the texts used by RS&V in gray.

| | GA | PWWS | PSO | Textfooler | HLA |
|---|---|---|---|---|---|
| N/A | 43.6 | 37.1 | 36.4 | 34.8 | 41.9 |
| FGWS | 48.3 | 45.4 | 42.7 | 50.7 | 50.9 |
| RS&V | **83.3** | **81.4** | **84.8** | **85.0** | **89.4** |

Table 4: The classification accuracy (%) of normally trained model, FGWS and RS&V on AG's News for Word-CNN in TAE setting.

further validate the effectiveness of RS&V, we consider a more rigorous setting of TAE, in which the attacker could access and attack the detection module to generate adversarial examples. Specifically, we generate adversarial examples on AG's News for Word-CNN in the TAE setting. Due to the high computational cost of DISP in inferring the extra models for the attacks, here we do not consider DISP as the detection baseline.

As shown in Table 4, FGWS exhibits little effectiveness against various attacks under the rigorous setting when compared with the normally trained model. One possible reason for the poor effectiveness of FGWS would be that the attacker could easily bypass the low-frequency words and generate adversarial examples by just using the high-frequency words. In contrast, RS&V could maintain the high effectiveness against various attacks. We postulate that the uncertainty of RS&V at each iteration makes it harder for the attacker to find good local minima to craft adversarial examples, leading to high robustness of RS&V. This further verifies the high effectiveness and superiority of RS&V.

## 4.4 ABLATION STUDY

We also conduct a series of ablation experiments to investigate the impact of hyper-parameters in RS&V, namely the substitution rate $p$ in the randomized substitution, the number of votes $k$ during the vote process, and the stopword

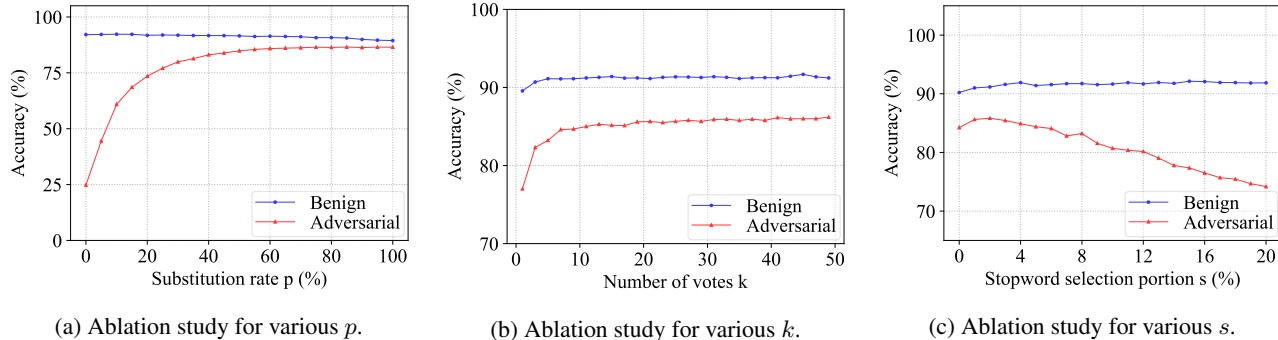

| (a) Ablation study for various $p$. | (b) Ablation study for various $k$. | (c) Ablation study for various $s$. |

Figure 3: The detection accuracy (%) of RS&V on AG's News dataset for Word-CNN against Textfooler, when varying the substitution rate $p$, number of votes $k$ or stopword selection portion $s$. The default values are $p = 60\%$, $k = 25$ and $s = 2\%$, respectively.

selection portion $s$. All the experiments are conducted on AG's News for Word-CNN against Textfooler using 1,000 randomly sampled texts that are correctly classified. To eliminate the variance of randomness, we repeat the experiments five times and report the average detection accuracy. The default values are $p = 60\%$, $k = 25$ and $s = 2\%$.

**On the effectiveness of substitution rate.** In Figure 3a, we study the impact of substitution rate $p$ on the detection accuracy. When $p = 0$, RS&V degenerates to the normally trained model without the detection module, which exhibits high classification accuracy on benign samples but low detection accuracy on adversarial examples. With the increment on the value of $p$, more words in the input text would be substituted, but the accuracy on the benign sample is stable with slight decay, indicating the model's high robustness against such randomized synonym substitution. In contrast, the detection accuracy on adversarial examples increases significantly when we increase the value of $p$ till $p = 60\%$, which is intuitive that replacing more words in the input text with synonyms would be more likely to eliminate the adversarial perturbation. Thus, we set $p = 60\%$ in the main experiments.

**On the effectiveness of number of votes.** As shown in Figure 3b, when the number of votes $k = 1$, there would be only a single input text and the vote cannot make any difference, hence, RS&V exhibits the lowest accuracy on either benign samples or adversarial examples. With the increment on the value of $k$, the accuracy on both benign samples and adversarial examples increases significantly, especially for the detection accuracy. With a large value of $k \geq 10$, RS&V maintains high and stable performance, supporting that our vote could eliminate the variance of randomized substitution and stabilize the detection process. In the main experiments, we simply select a large value of $k = 25$ on all datasets.

**On the effectiveness of stopword selection portion.** We continue to investigate the impact of the stopword selection portion $s$ on the detection accuracy. As illustrated in Fig-

ure 3c, the classification accuracy on benign samples is generally stable for various values of $s$, as our vote strategy has mitigated the variance introduced by randomness, leading to high and stable performance. The detection accuracy, however, increases when $s < 2\%$ and decreases significantly when more words in the input text cannot be substituted and thus could not eliminate the adversarial perturbation effectively. In the main experiments, we set $s = 2\%$ for better performance.

In summary, the substitution rate $p$, number of votes $k$ and stopword selection portion $s$ significantly influence the performance when they are small ($p \leq 60\%$, $k \leq 10$ or $s \leq 2\%$), while the performance becomes stable for larger $p$ and $k$ but decreases for larger $s$. In our experiments, we adopt $p = 60\%$, $k = 25$ and $s = 2\%$ for high and stable performance.

## 5 CONCLUSION

In this work, we identify that randomized synonym substitution could destroy the mutual interaction among words in the replacement sequence for adversarial attacks. Based on this observation, we propose a novel adversarial example detection method called Randomized Substitution and Vote (RS&V) to effectively detect the textual adversarial examples. RS&V adopts randomized synonym substitution to eliminate the adversarial perturbation and utilizes the accumulated votes to mitigate the variance introduced by randomness. Our method is generally applicable to any neural models without additional training or modification on the models. Empirical evaluations demonstrate that RS&V could achieve better detection performance no matter whether the attacker could access the detection module than existing baselines, at the same time RS&V maintains the high accuracy on benign samples. Moreover, RS&V identifies the fragility of textual adversarial examples, which might inspire more defense and detection methods by preprocessing the input text without sacrificing the classification accuracy on clean data.

## Acknowledgements

This work is supported by National Natural Science Foundation (62076105) and Hubei International Cooperation Foundation of China (2021EHB011).

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
