# OpenReview forum: "Detecting Textual Adversarial Examples through Randomized Substitution and Vote"
_auai.org/UAI/2022/Conference — UAI 2022 Poster_

### Official Review · Reviewer_PmjC · 2022-04-04

**Q2(1) Originality/Novelty:** 3
**Q2(2) Significance/Impact:** 3
**Q2(3) Correctness/Technical Quality:** 3
**Q2(6) Clarity Of Writing:** 3
**Q6 Overall Score:** 7
**Q8 Confidence In Your Score:** 3

**Q1 Summary And Contributions:**

This paper proposes RS&V, which detects textual adversarial examples by randomized substitution and vote. This method randomly substitutes some non-stopwords in a text with synonyms, generates multiple texts in this way, and calls the attacked classifier to get the predicted logits of each text. By comparing the accumulated logits of generated texts and the logits of original text, RS&V determines whether the original text is a benign example or an adversarial example.

**Q2 Assessment Of The Paper:**

More detailed information regarding each of these aspects is given below:

**Q2(4) Quality Of Experiments (Optional):**

3: Good: The experimental evaluation is adequate, and the results convincingly support the main claims.

**Q2(5) Reproducibility:**

3: Good: Key resources (e.g., proofs, code, data) are available and key details (e.g., proofs, experimental setup) are sufficiently well-described for competent researchers to confidently reproduce the main results.

**Q3 Main Strengths:**

1.	The motivation is reasonable. By substituting words with synonyms, successful attacks are rare, and most of the time the prediction results remain unchanged.
2.	The method is simple and effective, and the presentation of the method is clear.
3.	The experimental results show the effectiveness of the method.


**Q4 Main Weakness:**

1.	RS&V generates k=25 texts and queries the attacked model separately for one detection. Even for k=10 (a value to obtain stable performance), the cost is a bit high. In contrast, DISP and FGWS restore benign texts directly, which should be much faster.

**Q5 Detailed Comments To The Authors:**

See above

**Q7 Justification For Your Score:**

Novelty, Experimental results

**Q9 Complying With Reviewing Instructions:**

1: Yes.

---

### Official Review · Reviewer_6Ep9 · 2022-04-10

**Q2(1) Originality/Novelty:** 3
**Q2(2) Significance/Impact:** 3
**Q2(3) Correctness/Technical Quality:** 3
**Q2(6) Clarity Of Writing:** 4
**Q6 Overall Score:** 7
**Q8 Confidence In Your Score:** 4

**Q1 Summary And Contributions:**

The authors propose a new textual adversarial example detection method (RS&V). The method has the goal to contrast synonym substitution-based textual adversarial attacks by substituting random words in the text samples with the aim of destroying the mutual interaction between words. This interaction is used by attackers to trick the model and its destruction in adversarial samples improves the model’s accuracy.

**Q2 Assessment Of The Paper:**

More detailed information regarding each of these aspects is given below:

**Q2(4) Quality Of Experiments (Optional):**

3: Good: The experimental evaluation is adequate, and the results convincingly support the main claims.

**Q2(5) Reproducibility:**

2: Fair: Key resources (e.g., proofs, code, data) are unavailable but key details (e.g., proof sketches, experimental setup) are sufficiently well-described for an expert to confidently reproduce the main results.

**Q3 Main Strengths:**

- The paper is well written and the structure is crystal clear and helps to understand the idea and how it is implemented;
- the method is well explained both using a verbose description and an algorithmic one (Alg.1).
- the idea behind the method seems to be well justified by the experiments described in section 3.2 (with some doubts reported in the next section).
- the model is compared to two recent proposed methods (i.e. the baseline models);
- the method obtains very high performances compared to the baseline models.


**Q4 Main Weakness:**

The are only two main weakness:
-  that the code is not available;
- section 3: the motivation is based only on 1000 samples from one dataset. Why did you not tried on many samples from different datasets?



**Q5 Detailed Comments To The Authors:**

The authors propose a new textual adversarial example detection method (RS&V). The method has the goal to contrast synonym substitution-based textual adversarial attacks by substituting random words in the text samples with the aim of destroying the mutual interaction between words. This interaction is used by attackers to trick the model and its destruction in adversarial samples improves the model’s accuracy.

From a preliminary experiment seems clear that benign samples are not affected by the substitution of some words with their synonyms, while adversarial ones are hardly affected helping the model to correctly classify them. Starting from this analysis the authors propose this method. The method involves the creation of k variation of each sample and making the model predict a label for each variation. If the most predicted label is equal to the one predicted by the model for the original sample then the text was not corrupted, otherwise yes (the sample is an adversarial one).


The paper is well written and the proposed method is effective; it would be interesting to see it in other tasks. Moreover, I suggest you try to eliminate the s hyperparameter using a fixed set of stop-words (not dependent by the samples).

The are only two main weakness:
-  that the code is not available;
- section 3: the motivation is based only on 1000 samples from one dataset. Why did you not tried on many samples from different datasets?

Other some uncovered points:
- Section 3.2:  if you mask randomly, how do you know that is the break of mutual interaction of words that leads to these results? In the adversarial samples, maybe you masked the adversarial words and this increases the accuracy. A small analysis of this case would have been interesting.
- Last paragraph of section 3.2: Even in this case,  how could you say that the performance's improvement is due to substitution of surrounding words (respect to adversarial word) or of the adversarial word itself? Is clear that the probability of substituting a surrounding word is significantly higher,  but a more in-depth analysis would have been interesting.
- section 3.3: for the research of synonyms why not use a regex-based algorithm? In this case, you are almost sure to take a synonym, while using the embedding produced by a neural network could not be safe.
- section 3.3: p is a fixed number or a probability? in Alg. 1 and in the rest of the paper it seems a probability, and this makes sense, however here you say “we randomly sample p words”, this is probably a mistake;


**Q7 Justification For Your Score:**

The paper is well written and the proposed method is both simple and effective (looking at the results table). Some choices are not completely justified (like using similarity metrics of glove embedding instead of a rule-based method), but they seem to work very well. Moreover, the structure of the paper permits an immediate comprehension of the method.

**Q9 Complying With Reviewing Instructions:**

1: Yes.

---

### Official Review · Reviewer_PTHq · 2022-04-11

**Q2(1) Originality/Novelty:** 2
**Q2(2) Significance/Impact:** 3
**Q2(3) Correctness/Technical Quality:** 3
**Q2(6) Clarity Of Writing:** 3
**Q6 Overall Score:** 5
**Q8 Confidence In Your Score:** 3

**Q1 Summary And Contributions:**

This paper focuses on detecting textual adversarial examples and proposes a simple but effective method called RS&V against synonym substitution-based attack by breaking the mutual interaction of the words in the replacement sequence. The contribution is more applicative than theoretical.

**Q2 Assessment Of The Paper:**

More detailed information regarding each of these aspects is given below:

**Q2(4) Quality Of Experiments (Optional):**

2: Fair: The experimental evaluation is weak: important baselines are missing, or the results do not adequately support the main claims.

**Q2(5) Reproducibility:**

4: Excellent: Key resources (e.g., proofs, code, data) are available and key details (e.g., proof sketches, experimental setup) are comprehensively described for competent researchers to confidently and easily reproduce the main results.

**Q3 Main Strengths:**

1)	The motivation of the paper is clear. The whole manuscript is well-organized and easy to follow.
2)	The experiments are sufficient enough and validate the hypothesis at the beginning of the paper.
3)	The proposed method RS&V is simple but effective, which seems applicable to reality situation.



**Q4 Main Weakness:**

1)	The key idea of this paper is not innovative enough. The RS module only substitutes the word in the replacement sequence at the training phase randomly, which seems more like a data augmentation method. To this end, I suggest the authors move a step forward to give more theoretical explanations on how RS&V works rather than just using empirical experiments for validation. Besides, I am wondering whether this RS can be applied with other related tasks?
2)	Since every word needs a synonym set, the authors should evaluate the efficiency of this work.


**Q5 Detailed Comments To The Authors:**

Please refer to weaknesses.


**Q7 Justification For Your Score:**

This paper gives clear motivation and well-organized manuscript. The experiments are sufficient and the results are convincing. However, the proposed method is not innovative enough. I will give a ‘weak accept’ to this paper.

**Q9 Complying With Reviewing Instructions:**

1: Yes.

---

### Official Review · Reviewer_HRVr · 2022-04-13

**Q2(1) Originality/Novelty:** 1
**Q2(2) Significance/Impact:** 1
**Q2(3) Correctness/Technical Quality:** 2
**Q2(6) Clarity Of Writing:** 2
**Q6 Overall Score:** 3
**Q8 Confidence In Your Score:** 5

**Q1 Summary And Contributions:**

This work designs a randomized substitution-based approach to detect textual adversarial examples. The proposed approach could be integrated with other models without further modification which is the merit. Experiments are evaluated on three datasets with good results.




**Q2 Assessment Of The Paper:**

More detailed information regarding each of these aspects is given below:

**Q2(4) Quality Of Experiments (Optional):**

2: Fair: The experimental evaluation is weak: important baselines are missing, or the results do not adequately support the main claims.

**Q2(5) Reproducibility:**

2: Fair: Key resources (e.g., proofs, code, data) are unavailable but key details (e.g., proof sketches, experimental setup) are sufficiently well-described for an expert to confidently reproduce the main results.

**Q3 Main Strengths:**

1.	This paper designs a randomized substitution-based approach model to detect textual adversarial examples.
2.	A number of experiments have been performed to evaluate the proposed model.



**Q4 Main Weakness:**

1.	This paper is not well motivated as there exists a good number of related work, why you propose this randomized synonym substitution-based approach? What is the relationship between this work and the literature.
2.	It is hard to figure out a meaningful real-world application which needs NLP defense against adversarial examples.
3.	The compared methods are not strong enough and thus the results are not convincing.



**Q5 Detailed Comments To The Authors:**

It is suggested to start with a more reasonable example to demonstrate the necessity of the proposed work. The related work should be well discussed to show the contribution of this manuscript. Experiments should be greatly enhanced.

**Q7 Justification For Your Score:**

The paper is not well motivated and it is hard to understand why we need this work.

**Q9 Complying With Reviewing Instructions:**

1: Yes.

---

### Decision · Program_Chairs · 2022-05-15

**Decision:**

Accept (Poster)

**Comment:**

Meta Review: For the most part reviewers felt the proposed method was proven to be effective and that the paper constitutes a useful contribution.